# Consumption Upgrading and Industrial Structural Change: A General Equilibrium Analysis and Empirical Test with Low-Carbon Green Transition Constraints

**Xiaowei Xing [1],\* and Azhong Ye [2]**

[1] School of Business Administration, Northeastern University, Shenyang 110167, China
[2] School of Economics and Management, Fuzhou University, Fuzhou 350108, China
\* Correspondence: 2010444@stu.neu.edu.cn

**Abstract:** To clarify the relations between low-carbon green transition, consumption upgrading, and industrial structure change, this paper firstly builds a dynamic model of the three, then uses the PVAR Model and panel data of 30 Chinese provinces from 2008 to 2020 to carry out empirical study from the rationalization and upgrading dimensions of industrial structural change, respectively. The results are as follows: (1) Low-carbon green transition and consumption upgrading are Granger causes of each other. In this causal relationship, low-carbon green transition hinders consumption upgrading, but consumption upgrading significantly promotes low-carbon green transition. (2) Low-carbon green transition plays a facilitating and hindering role in industrial structure rationalization and upgrading, respectively. However, from the different dimensions of industrial structure change, only industrial structure upgrading has a significant reverse hindering effect on low-carbon green transition, and the reverse effect of industrial structure rationalization on low-carbon green transition is not significant. (3) Consumption upgrading has a hindering and promoting effect on the rationalization of industrial structure in the short- and long-run respectively, and a promoting and hindering effect on the industrial structure upgrading respectively; however, only industrial structure upgrading significantly promotes consumption upgrading in the opposite direction, while industrial structure rationalization has no significant effect on consumption upgrading. These findings propose some suggestions such as advocating the new way of green consumption, constructing and improving the green whole industry chain, and strengthening the synergy between imitative innovation and independent innovation.

**Keywords:** green transition; consumption upgrading; industrial structural change; general equilibrium model; dynamic interaction effects

## 1. Introduction and Literature Review

To address global climate change and fulfill its role as a great power, China has been implementing green and innovative development transformation initiatives, proposing the "3060" development target for carbon emissions and incorporating carbon peaking and carbon neutrality into the overall layout of ecological civilization construction. Consumption, as an important cornerstone of sustainable economic development, will also be affected by the green transformation in the context of carbon neutrality and carbon peaking. This will occur especially as consumption concepts and patterns change and green consumption is practiced, with people pursuing healthier, greener, and more ecological products of higher quality, prompting enterprises to take a deeper leap in production and services and reshape the original industrial structure and production methods. In addition, to break through the bottlenecks of resources and the environment and respond to the trend of consumption upgrade, it is necessary to develop and foster green and low-carbon industries, promote the green and low-carbon transition of the stock industries, transform the traditional economic growth model by industrial transformation and upgrading, and promote green and low-carbon high-quality development of the economy and society.

However, due to the insufficient and unbalanced economic and social development caused by the institutional changes during China's economic transition, the relationship between low-carbon green transition, consumption upgrading and industrial structure change has not yet formed a unified view, and there is a certain degree of controversy, which in turn is not conducive to the promotion of the construction of ecological civilization and the implementation of green and sustainable development strategies in China. So, what are the intrinsic mechanisms of low-carbon green transition, consumption upgrading and industrial structural change in China? What are the effects of each other, facilitating or inhibiting? This paper attempts to analyze the above questions through a combination of theoretical analysis and empirical research, with a view to clarifying the relationship between low-carbon green transition, consumption upgrading and industrial structure change in China's current economic development process, which is important for China to better implement green sustainable development policies, consumption upgrading promotion policies and industrial structure adjustment policies, and to achieve the coordinated development of green transition, industrial supply and consumption expansion and upgrading.

The remainder of this study is structured as follows: Section 2 is the literature review section; Section 3 theoretically identifies the interactive effects of consumption upgrading and industrial structure changes under the constraints of low-carbon green transition; Section 4 introduces the construction of the econometric model and empirical research data; Section 5 provides the empirical analysis of the interactive relationship between low-carbon green transition, consumption upgrading, and industrial structure change; and Section 6 presents the research conclusions and policy recommendations.

## 2. Literature Review

Many scholars have already studied the two-by-two relationship between green transition, resident consumption upgrading, and industrial structural change, but there is not much literature on the dynamic relationship between the three variables. Extant studies primarily focus on the following aspects:

First, the relationship between the low-carbon green transition and people's consumption. As an important outgrowth of ecological civilization, green development tends to positively empower consumption and contributes to the accelerated formation of green consumption, which has a positive impact on consumption upgrading. However, green development may also lead to a shortage of supply because it forces enterprises to change their production models to meet the increasingly diversified and personalised demand for green products, which has a crowding-out effect on residents' consumption and affects changes in consumption structure [1], and the degree of consumer satisfaction in consumption activities depends on the carrying capacity of resources and the environment [2]. The excessive expansion and upgrading of consumption will also put certain pressure on the environment and even cause waste and consumption of resources [3]. In addition, some scholars argue that green development in China has a differential impact on the expansion of consumption by different income groups, with the expansion of consumption by high-income and low-income residents having a facilitating and inhibiting effect, respectively [4].

The second aspect is the change in residents' consumption and industrial structure, mainly from the one-way and interactive influence relationship between industrial structure and residents' consumption. (1) The influence of industrial structure on residents' consumption: researchers generally believe that the upgrading of industrial structure as a whole can promote the upgrading of residents' consumption [5]; specifically, the independent innovation of industry can improve the quality of supply and fill the supply gap, promoting the growth and upgrading of consumption [6], although this influence effect has regional and urban-rural heterogeneity. (2) The influence of residents' consumption on the industrial structure: academia has reached a consensus that consumption demand guides the direction of development of production and promotes industrial structure optimiza-

tion by influencing production structure [7–9], but the lag in optimizing the consumption structure of low-income groups can lead to a less pronounced role in promoting industrial structure [10]. (3) The interactive effect of industrial structure and residents' consumption: some scholars believe that industrial structure and consumption expansion and upgrading are complementary, that industrial upgrading must be compatible with consumption upgrading, and that consumption upgrading is a measure of industrial upgrading [11]. However, regarding the strength of their mutual influence, Zhao and Wang argue that the impact of industrial structure transformation and upgrading on changes in residents' consumption structure is greater than the impact of changes in residents' consumption structure on industrial structure transformation and upgrading [12], whereas Cai argues that industrial structure upgrading has little impact on the increase in residents' consumption [13], Lai et al. argue that industrial structure upgrading will not promote consumption upgrading, and may even inhibit it in some cases [14].

The third aspect is low-carbon green transition and industrial structure. Low-carbon green transition may inhibit technological innovation, and when environmental policies are tightened, the increased environmental costs of enterprises may crowd out investment in technological innovation, which is not conducive to technological innovation; however, on the other hand, low-carbon green transition can improve the efficiency of enterprises' resource use, effectively force them to update their production technology and promote both a circular economy and low-carbon development, which not only helps to improve their production efficiency and market share, but can also promote the optimisation and upgrading of industrial structure [15,16]. As China implements differential environmental policies across regions and industries, it is prone to inter-regional migration of firms and industries, which also affects inter-regional industrial structure changes [15,17]. In addition, some scholars argue that the impact of low-carbon green transition on an industrial structure depends on the nature of environmental regulation [18]. Compared to the promotion effect of formal environmental regulation, Wang et al. argue that informal environmental regulation promotes industrial structure upgrading in the early stage and inhibits it in the middle and late stages [19].

In addition, some of the literature has also examined the interaction between low-carbon green transition, consumption upgrading and industrial structure change in China. For example, Zou and Wu construct a DSGE model and use numerical simulation to find that the positive effect of improving energy use efficiency and reducing environmental pollution on China's economic development is increasing. It is imperative to reduce the capital margin, eliminate inefficient production capacity, and achieve industrial transformation. Although industrial transformation may impact old industries, leading to the rapid decline of some traditional industries, causing large fluctuations in employment and income, and will not be conducive to improving the consumption structure [20].

Through reviewing the domestic and international literature, it is found that many scholars have made important contributions to the study of low-carbon green transition, residents' consumption upgrading, and industrial structure change. Even so, most of them only analyze the relationship between the two and ignore the endogeneity caused by the inner interactions. Few studies include low-carbon green transition, consumption upgrading, and industrial structural change in China in the same research framework and examine the linkage effects. In this study, based on previous studies, we analyze the influence mechanism between the three in a unified dynamic system by constructing a theoretical model of consumption upgrading and industrial structure change under the constraint of low-carbon green transition and empirically analyze the dynamic relationship between the three using the Panel Vector Autoregressive Model (PVAR) model that reveals the interaction effects of the variables, to provide theoretical and empirical support for the formulation of relevant policies and the construction of a new development pattern.

The marginal contributions of this paper are as follows: First, by introducing low-carbon green transition into a three-sector dynamic general equilibrium model, this paper theoretically elaborates a mechanism for examining the interaction between consumption

upgrading and industrial structure change under the constraint of low-carbon green transition, enriching the relevant literature. Second, considering both subjective and objective factors in the process of consumption upgrading, this paper constructs a comprehensive evaluation index system for consumption upgrading that more comprehensively and objectively reflects the level of consumption upgrading. Third, this paper analyzes the causal relationship between three factors using Granger causality testing and the impulse response function of the PVAR model and evaluates the strength of the interaction between the three factors using variance decomposition, which is useful for clarifying the interaction mechanism between low-carbon green transition, consumption upgrading, and industrial structure change and will help the Chinese government formulate and improve reasonable and scientific green sustainable development policies.

## 3. Theoretical Modeling and Derivation

Building on Ngai and Pissarides [21], Alvarez-Cuadrado and Poschke [22], Sun and Xu [6], this study introduces carbon emission allowances related to low-carbon green transition into a three-sector dynamic general equilibrium model to examine the consumption upgrading and industrial structure under the constraints of low-carbon green transition in China.

### 3.1. Enterprise Production

Assume that there are two main production sectors: basic goods and desired goods. Assume that the two production sectors satisfy the following: (i) the labour supply is inelastic, and the total labour unitisation is 1, that is, $L_t = L_t^B + L_t^A = 1$ where $L_t^B$ and $L_t^A$ are the number of labourers in the basic and desired goods sectors, respectively, in the $t$ period. (ii) The output of the basic goods sector $Y_t^B$ is used entirely for final consumption $(C_t^B = Y_t^B)$. (iii) A portion of the desired product output $Y_t^A$ is used for final consumption $C_t^A$, and the remainder is used for capital accumulation $K_t^A$, that is, $C_t^A + K_t^A = Y_t^A$. (iv) Factor markets are perfectly competitive, and labour and capital are free to move between the two sectors.

The production functions for each of the two sectors that incorporate Hicks' neutral technological progress are set as follows:

$$\begin{cases} Y_t^B = (A_t^A)^\theta (A_t^B)^{1-\theta} (L_t^B)^\alpha (K_t^B)^{1-\alpha} \\ Y_t^A = A_t^A (L_t^A)^\alpha (K_t^A)^{1-\alpha} \end{cases} \tag{1}$$

In Equation (1), $Y_t^B$ and $Y_t^A$ are the product outputs of the basic and desired sectors, respectively; $A_t^A$ and $A_t^B$ are the levels of accumulation of frontier technologies for autonomous innovation capabilities and technology absorption in the basic product sector, respectively; and $\theta$ reflects the spillover of frontier technologies to the basic sector ($\theta < 1$). Assuming that $A_t^A > A_t^B$, that is, the level of technology absorption in the basic sector is strictly below the level of technological innovation in the desired sector, there is $(A_t^A)^\theta (A_t^B)^{1-\theta} < A_t^A$.

In addition, since the enterprise's production will also generate pollution emissions, and because the only way to reduce negative externalities is to control pollution emissions, pollutant emission limits are a 'bad' input to the production process. Assuming that the homogeneous carbon emitted by the production process in both sectors constitutes the only source of ecological pollution and that carbon emissions from both sectors are limited by carbon emission allowances, the two sectoral production functions are extended as follows:

$$\begin{cases} Y_t^B = [(A_t^A)^\theta (A_t^B)^{1-\theta} (L_t^B)^\alpha (K_t^B)^{1-\alpha}](S_t^B)^\varphi \\ Y_t^A = [A_t^A (L_t^A)^\alpha (K_t^A)^{1-\alpha}](S_t^A)^\varphi \end{cases} \tag{2}$$

In Equation (2), $S_t^A$ and $S_t^B$ are the carbon allowances for the base and desired sectors, respectively; $\varphi$ reflects the output elasticity of carbon allowances; and $\varphi \in [0,1]$. Both sectors maximise their profits by choosing the amount of capital and labour subject to the carbon emission allowance constraint.

### 3.2. Consumer Utility

Assuming the existence of a large number of homogeneous households capable of surviving indefinitely and drawing on Kongsamut et al.'s setting of consumption preferences for different sectoral products [23], the lifetime and instantaneous utility functions for households are set as follows:

$$V(C^B, C^A) = \int_0^\infty e^{-\rho t} \ln[U(C^B, C^A)]dt \tag{3}$$

$$U(C^B, C^A) = [\xi^{1/\varepsilon}(C_t^B - \mu)^{(\varepsilon-1)/\varepsilon} + (1-\xi)^{1/\varepsilon}(C_t^A)^{(\varepsilon-1)/\varepsilon}]^{\varepsilon/(\varepsilon-1)} \tag{4}$$

where Equation (3) is the lifetime utility function, Equation (4) is the instantaneous utility function, and the household will choose between the base good ($C^B$) and the desired good ($C^I$) to maximise intertemporal utility. In Equation (3), $\rho$ is the time preference rate; in Equation (4), $\xi \in [0,1]$ is the consumer preference for the basic good, with a larger $\xi$ representing a greater preference for the basic good; $\varepsilon$ is the elasticity of substitution between the two goods: $\varepsilon > 1$ indicates that the basic good and the desired good are substitutes for each other and $0 < \varepsilon < 1$ that they are complements; $\mu$ is the minimum share of food expenditure to sustain the household, with a value greater than a zero constant and $\mu < C_t^B$ ensuring a constant positive labour force in the basic sector ($L_t^B > 0$). $\mu$ can also indicate that the income elasticity of the basic product is less than one, a setting that is consistent with realistic consumption behaviour. In line with Kongsamut et al. [23], this study assumes that the income elasticity of a desired product is equal to one.

The household income is derived from wage income, capital gains, and government transfers in two main sectors and uses the resulting income to choose to buy two products or accumulate capital; the household budget constraint in general equilibrium is expressed as follows:

$$\omega_t L_t^B + \omega_t L_t^A + r_t K_t^B + r_t K_t^A - \delta K_t^A + G_t = C_t^B + P_t C_t^A + I_t \tag{5}$$

where $\omega_t$ and $r_t$ are the level of wages and capital gains, respectively; $G_t$ is the one-off government transfer to households; $P_t$ is the relative price of the desired product relative to the base product; and $\delta$ is the capital depreciation rate.

### 3.3. Government Departments

Assume that the government sets the total current carbon emissions at the beginning of each period based on the need for green and low-carbon transition $S_t$ ($0 < S_t < 1$, the total carbon emissions that the natural environment can accommodate is normalised to 1 in this study under the premise of ensuring human survival) and the price per unit of carbon emissions, and sets the current carbon emission allowances for both sectors. In addition, assuming that the government transfers all the expected carbon emission revenues to consumers at the beginning of the period, thus returning the distortionary taxes collected from economic agents to the economy through transfer payments and ensuring that its budget is balanced, there are:

$$\begin{cases} G_t = \tau_t S_t^B + \tau_t S_t^I \\ S_t = S_t^B + S_t^I \\ 1 - S_t > 0 \end{cases} \tag{6}$$

In Equation (6), $G_t$ is the government's carbon revenue, $\tau_t$ is the unit carbon price, $S_t$ is the total carbon emissions, and $S_t^A$ and $S_t^B$ are the carbon allowances for the base and desired sectors, respectively.

*3.4. Model Solving and Analysis*

3.4.1. Business Production and Consumer Upgrading

Under factor-free arbitrage conditions, the wage level and capital gains of labour in both sectors are equal to the marginal output of the two factors in each sector. If the price of the underlying product is normalised to 1, the price of the desired product relative to the price of the underlying product $P_t$ satisfies

$$P_t = \left(\frac{A_t^B}{A_t^A}\right)^{1-\theta} \left(\frac{S_t^B}{S_t^A}\right)^{\varphi} \tag{7}$$

Equation (7) shows that the price of the desired product relative to the base product is influenced by the technological (absorption) capacity of both sectors and the carbon emission quotas. If the desired sector's independent innovation capacity increases or has higher carbon emission allowances, the relative price of the desired product $P_t$ will tend to decrease, which will increase the consumption of the desired product and promote the upgrading of both the consumption structure and industrial structure. Conversely, the relative price of the desired product will increase, leading to an inverse advanced transformation of the consumption structure and industrial structure.

From Equation (7) and the Lagrangian first order condition, it follows that

$$C_{stru,t} = \frac{C_t^A}{C_t^B - \mu} = \left(\frac{1-\xi}{\xi}\right)\left[\left(\frac{A_t^A}{A_t^B}\right)^{1-\theta}\left(\frac{S_t^A}{S_t^B}\right)^{\varphi}\right]^{\varepsilon} \tag{8}$$

The middle part of Equation (8) shows that the consumption structure escalates as the ratio of the total consumption of the desired product ($C_t^A$) to the consumption of the basic product (net of minimum food expenditure) ($C_t^B - \mu$). The rightmost part of the equation reflects the total impact of technological (absorptive) capacity and carbon emission quotas, consumption preferences, and so on, on the consumption structure. If the desired product sector has a higher capacity for autonomous innovation or carbon emission quotas, its optimal output $Y_t^A$ will increase ((Equation (2)). When the desired product has a comparative price advantage, consumers will increase the quantity of the desired product ($C_t^A$) consumed, and consumers' increased preference for the desired product ($1 - \xi$) will also increase the quantity of the desired product ($C_t^A$), promoting the upgrading of the consumption structure.

3.4.2. Consumption Upgrading and Factor Allocation

Equation (8) and the factor no-arbitrage condition led to the equation for total household consumption (containing consumption of the desired product and base product) and total output.

$$Y_{real,t} = (1/P_t)Y_t^B + Y_t^A = A_t^A L_t^\alpha K_t^{1-\alpha}(S_t^A)^{\varphi} \tag{9}$$

$$C_{real,t} = (1/P_t)C_t^B + C_t^A \text{ that is } (C_t - \mu)_{real,t} = C_t^A X \tag{10}$$

where in Equation (10), $X = x + 1$, $x = P_t^{\varepsilon-1}[\xi/(1-\xi)]$. Equations (9) and (10) show that the real value of total output ($Y_{real,t}$) depends on the desired sectoral frontier level of technological accumulation ($A_t^A$) and carbon emission allowances ($S_t^A$), whereas the real value of total consumption after deducting people's minimum subsistence food needs expenditure ($(C_t - \mu)_{real,t}$) is related only to the total desired product consumption ($C_t^A$).

Setting the labour share of the basic sector $SL^B$ to the following form [21], which can be derived from Equations (2), (3), (7)–(9) and (10).

$$SL_{real}^B = \frac{Y_{real}^b}{Y_{real}} = \frac{C_{real}^b}{Y_{real}} = \frac{(C-\mu)_{real}}{Y_{real}} \times \frac{x}{X} + \frac{\mu_{real}}{Y_{real}} \tag{11}$$

Equation (14) is the interaction effect between the efficiency of labour factor allocation and consumption in the general equilibrium. According to Equation (14), (i) other things being equal, the ratio of total consumption to total output is fixed ($(C - \mu)_{real}/Y_{real}$ is a constant), and the share of labour in the basic sector $SL^B_{real}$ decreases as the total output $Y_{real}$ increases. (ii) From $\partial SL^B_{real}/\partial x > 0$; if the share of consumer spending on basic goods ($x$) increases, labour flows to the basic goods sector. Conversely, labour flows to the desired goods sector as consumer spending on desired goods increases.

## 4. Empirical Design

### 4.1. PVAR Model and Estimation Method

Holtz-Eakin [24] proposed the PVAR model, which has been continuously expanded by scholars, such as Love et al. [25], and has been used in a wide range of economics research. The PVAR model is not based on any a priori economic theory; it treats all variables as endogenous, allows for interactions between variables, and combines the strengths of panel and VAR models to systematically analyze the dynamic responses of economic variables to shocks while controlling for regional individual and time effects. It provides an effective test for a more realistic and comprehensive reflection of the dynamic relationship between low-carbon green transition, consumption upgrading, and industrial structure change, and its general form is

$$Y_{it} = \alpha_0 + \beta_0 + \sum_{j=1}^{p} \beta_j Y_{i,t-j} + \alpha_i + \gamma_t + \mu_{it} \tag{12}$$

The subscripts $i$ and $t$ represent sample province $i$ and year $t$, respectively, $Y$ is a vector containing the variables carbon neutral and green transformation, consumption upgrading and industrial structure change, $p$ represents the optimal lag of the PVAR model, $\alpha_0$, $\alpha_i$, $\gamma_t$ and $\mu_{it}$ are the intercept, individual effect, time effect, and random disturbance terms, respectively, and $\beta_j$ is the regression coefficient matrix.

As the explanatory vector of the PVAR model contains lagged terms for endogenous variables and individual heterogeneity due to individual (time) effects, it has a similar econometric test to the dynamic panel model, which requires that the endogeneity of variables and individual (time) effects are dealt with effectively before the model is estimated. To achieve this, the data are first subjected to a "Helmert process" to remove sample time and individual fixed effects, ensure that the transformed variables are orthogonal to the lagged variables and independent of the random disturbance terms, and finally estimate the parameters of the PVAR model using a generalized method of moments (GMM) estimation with the lagged variables as instrumental variables. (Based on the program provided by Lian Yujun and Love).

### 4.2. Sample Data

This study uses annual data from 30 provinces, cities, and autonomous regions in mainland China (excluding the Tibet Autonomous Region due to missing data) for 2008–2020, with a total of 390 research samples. The data for each indicator are obtained from the 2009–2021 China Statistical Yearbook, China Energy Statistical Yearbook, provincial statistical yearbooks, the website of the National Bureau of Statistics, with some missing data supplemented by linear interpolation or extrapolation.

#### 4.2.1. Low-Carbon Green Transition (CNGT)

Reducing carbon emissions to achieve a green transition is a practical action to address climate change, and a green transition can be achieved by practicing low-carbon production and lifestyle. This study used carbon-neutral indicators to quantify the low-carbon green transition based on energy consumption data from the China Energy Statistics Yearbook, and the China Carbon Accounting Database emissions are summed up. The carbon intensity

is expressed as the ratio of carbon emissions to the gross domestic product (GDP; unit: 10,000 tons/billion yuan) and is calculated as follows:

$$CO_{2it} = [\sum (E_{jit} \times NCV_j \times CC_j \times COF_j \times \frac{44}{12})]/GDP_{it} \tag{13}$$

where $CO_2$ is the carbon emission intensity of the province $i$ in the year $t$, $E_j$ is the consumption of various energy sources, $NCV_j$ is the average low level of heat generation of the energy source $j$, $CC_j$ is the carbon content per unit of heat, $COF_j$ is the oxidation factor of the energy source $j$, 44 and 12 are the carbon dioxide and carbon molecular weight, respectively, and $GDP$ is the gross domestic product of each region in a calendar year. The product of the three terms $NCV_j$, $CC_j$ and $COF_j$ represents the carbon emission factor, then the emission factor of $CO_2$ is $NCV_j \times CC_j \times COF_j \times 44/12$, as shown in Table 1.

**Table 1.** Carbon emission factors by type of energy consumed.

|  | Coal | Coke | Crude Oil | Fuel Oil | Petrol | Paraffin | Diesel | Natural Gas |
|---|---|---|---|---|---|---|---|---|
| Carbon emission factor | 0.7559 | 0.855 | 0.5538 | 0.5857 | 0.5921 | 0.5714 | 0.6185 | 0.4483 |

Data source: 2006 IPCC Guidelines for National Greenhouse Gas Emissions Inventories.

### 4.2.2. Consumption Upgrading (CSU)

In this study, we used the consumption upgrading index as a proxy variable for the upgrading of consumption in each region, following the studies of Du [26] and Ye [27]. We considered the changes in the structure of consumption objects, such as "Material-Services (Spiritual)" consumption or the transformation of subsistence demand to development and enjoyment demand, as well as the changes in consumption patterns and concepts in the process of upgrading consumption. The system is based on the principles of systematicity and scientificity and a comprehensive evaluation index system comprising 26 indicators (Table 2) in five dimensions: total consumption, consumption level, consumption content, consumption pattern, and consumption philosophy, and are integrated into a single comprehensive index of consumption upgrading using the entropy value method, to comprehensively reflect the comprehensive upgrading of consumption objects, consumption level, consumption pattern and philosophy of consumption subjects. The index is designed to reflect the overall upgrading of consumption objects, consumption levels, consumption patterns, and concepts of consumption subjects.

### 4.2.3. Industrial Structure Change

Industrial structural change is the reallocation of factors of production across economic sectors and different industries [28]. It is ultimately a convergent development toward the rationalization and modernization of industrial structure. Therefore, this study split industrial structural change into two dimensions: industrial structure rationalization ($TL$) and industrial structure upgrading ($IND$). Among them, industrial structure rationalization should reflect the degree of optimal allocation of each resource within an industry and the coordination of the proportional structure between industries. This study referred to Gan et al. [29], which measured industrial structure rationalization based on the generalized entropy method.

$$TL_t = \sum_{i=1}^{3} \left[ \left( \frac{Y_{it}}{Y_t} \right) \ln \left( \frac{Y_{it}}{L_{it}} \middle/ \frac{Y_t}{L_t} \right) \right] \tag{14}$$

In Equation (14), $TL_t$ is the degree of industrial structure rationalization in the year $t$, $Y_{it}$ is the total output of the industry $i$ ($i = 1, 2, 3$) in the year $t$, and $L_{it}$ is the number of employees in the industry $i$ in the year $t$. Equation (11) shows that in the process of measuring the rationalization of industrial structure, the three major industries are weighted by the proportion of output value to the total output value, effectively transforming them

into relative indicators based on the theory of structural deviation. The smaller the value of *TL*, the higher the degree of industrial structure rationalization; conversely, the lower the degree of industrial structure rationalization.

**Table 2.** Comprehensive evaluation index system for consumer upgrading.

| Guideline Level | Specific Indicators | Indicator Attributes | Indicator Weights | Guideline Level | Specific Indicators | Indicator Attributes | Indicator Weights |
|---|---|---|---|---|---|---|---|
| Overall social consumption | Consumption rate | + | 0.019 | Consumer content | Per capita consumption expenditure on household equipment and services | + | 0.031 |
| | Total social consumption | + | 0.065 | | Per capita consumption expenditure on transport and communications | + | 0.043 |
| | Growth rate of total social consumption | + | 0.003 | | Health care consumption expenditure per capita | + | 0.027 |
| | Number of workers in the tertiary sector | + | 0.018 | | Per capita consumption expenditure on education, culture, and entertainment | + | 0.033 |
| Consumption patterns | Total postal and telecommunications services | + | 0.008 | | Other consumption expenditure per capita | + | 0.037 |
| | Total express delivery per capita | + | 0.194 | | Developmental consumption as a percentage | + | 0.018 |
| | Telephone penetration rate | + | 0.024 | | Percentage of consumption for enjoyment | + | 0.008 |
| | Total restaurant and accommodation business | + | 0.088 | | Consumer Upgrades | + | 0.007 |
| | Service Levels in Catering and Accommodation | + | 0.080 | | Engel's coefficient | − | 0.011 |
| | Number of travel agents | + | 0.092 | | Car ownership | + | 0.038 |
| Consumption level of the population | Per capita consumption expenditure | + | 0.036 | Consumer Philosophy | Low Carbon Consumption | + | 0.003 |
| | Consumption growth rate | + | 0.007 | | Number of public transport rides per capita | + | 0.045 |
| | Urban to rural consumption ratio | − | 0.012 | | Risk management awareness | + | 0.052 |

Note: Due to space constraints, the formulae (methods) for calculating specific indicators are not reported and are kept on file for reference.

In accordance with the "Paddy-Clark theorem," the industrial structure will be upgraded from low to high level, with the transition from primary to secondary and then to tertiary industries; and the industries will move from the low end to the high end of the value chain, from low value-added to high value-added product sectors. For this reason, this study draws on the practice of He et al. [30] and others to measure the industrial structure upgrading (IND) by the ratio of tertiary to secondary output, with a larger IND value indicating a higher level of industrial structure upgrading.

## 5. Empirical Study

### 5.1. Smoothness Test and Choice of Optimal Lag Order

Before testing the dynamic relationship between variables through the PVAR model, unit root tests should be conducted on each variable to avoid the phenomenon of "Pseudo-regression". The results of this study are shown in Table 3, based on both the homogeneous *LLC* test and the heterogeneous *Fisher* − *ADF* test for the panel data: *CNGT*, *CSU*, *TL* and *IND* all reject the original hypothesis of a unit root at least at the 5% significance level, which means that the variables are smooth series. There may be a stable and long-term equilibrium relationship between the variables.

**Table 3.** Results of panel data stationarity tests.

| Variables | LLC Test | Fisher ADF Test | Conclusion |
|---|---|---|---|
| CNGT | −12.353 *** | 119.173 *** | Stable |
| CSU | −2.484 *** | 82.786 ** | Stable |
| TL | −17.629 *** | 242.334 *** | Stable |
| IND | −9.478 *** | 87.009 ** | Stable |

Note: **, *** denote significant levels of 5% and 1%, respectively.

Given the influence of the order of variables on the results of the PVAR model, the order of variables in the empirical model is set to *CNGT*, *CSU*, *TL* (*IND*), according to the existing research and the logic of this study. As for the optimal lag order of the PVAR model, comparing the values of the AIC criterion, BIC criterion, and HQIC criterion of the PVAR model with lags 1–5 in Table 4. It can be seen that the lag 1 PVAR model is the optimal choice when analyzed from the perspective of industrial structure rationalization or industrial structure upgrading. Therefore, this study eventually builds lagged 1st order PVAR models to analyze the interaction between *CNGT*, *CSU* and *TL* or *IND*, respectively.

**Table 4.** Optimal lag order judgment results.

| Lagging Order | TL Angle | | | IND Angle | | |
|---|---|---|---|---|---|---|
| | AIC | BIC | HQIC | AIC | BIC | HQIC |
| 1 | −9.221 * | −8.081 * | −8.767 * | −8.119 | −6.979 * | −7.664 * |
| 2 | −9.214 | −7.881 | −8.680 | −8.067 | −6.733 | −7.533 |
| 3 | −8.630 | −7.071 | −8.004 | −8.115 | −6.555 | −7.488 |
| 4 | −2.359 | −0.532 | −1.623 | −8.169 * | −6.341 | −7.432 |
| 5 | −7.753 | −5.601 | −6.883 | −3.850 | −1.698 | −2.980 |

Note: * denotes the optimal lag order.

### 5.2. Granger Causality Test

The Granger causality test also needed to verify whether the long-term equilibrium relationship between the variables constitutes a causal relationship. The results of the Granger causality test with a lag of 1 period are shown in Table 5. From the perspective of industrial structure rationalization, low-carbon green transition and consumption upgrading are Granger causes of each other at the 1% significance level. In comparison, low carbon green transition and consumption upgrading are one-way Granger causes of industrial structure rationalization at the 5% and 1% significance levels, respectively. This indicates that there is an interactive relationship between low-carbon green transition and consumption upgrading. Low-carbon green transition and consumption upgrading constitute Granger causes of industrial structure rationalization, but it is difficult to judge whether industrial structure rationalization constitutes Granger causes of low-carbon green transition and consumption upgrading based solely on the results in Table 5. In the industrial structure upgrading model, there is a two-way Granger causal relationship between low-carbon green transition, consumption upgrading, and industrial structure upgrading, indicating that there is an obvious dynamic interaction effect between low-carbon green transition, consumption upgrading, and industrial structure upgrading. The Granger causality test only provides the statistical significance of causality and can therefore be used as important evidence for the true causality between variables. However, the true interaction between the variables must be further tested based on the estimated results of the PVAR model.

**Table 5.** Granger causality test results.

| Variables | TL Model | | | Variables | IND Model | | |
|---|---|---|---|---|---|---|---|
| | Original Assumptions | Cardinality Test Value | Conclusion | | Original Assumptions | Cardinality Test Value | Conclusion |
| CNGT | CSU is not the reason | 6.797 *** | Rejection | CNGT | CSU is not the reason | 4.632 ** | Rejection |
| | TL is not the reason | 0.356 | Acceptance | | IND is not the reason | 2.992 * | Rejection |
| | ALL is not the reason | 12.301 *** | Rejection | | ALL is not the reason | 9.281 *** | Rejection |
| CSU | GNGT is not the reason | 7.808 *** | Rejection | CSU | GNGT is not the reason | 10.352 *** | Rejection |
| | TL is not the reason | 1.145 | Acceptance | | IND is not the reason | 4.194 ** | Rejection |
| | ALL is not the reason | 9.838 *** | Rejection | | ALL is not the reason | 10.474 *** | Rejection |
| TL | GNGT is not the reason | 3.881 ** | Rejection | IND | GNGT is not the reason | 11.421 *** | Rejection |
| | CSU is not the reason | 11.806 *** | Rejection | | CSU is not the reason | 5.260 ** | Rejection |
| | ALL is not the reason | 14.873 *** | Rejection | | ALL is not the reason | 13.945 *** | Rejection |

Note: *, **, *** denote significant levels of 10%, 5% and 1%, respectively.

### 5.3. PVAR Model GMM Parameter Estimation

As an extended form of the VAR model, the PVAR model is a spent theoretical model, and the economic interpretation of its parameter estimates does not have much practical significance; that is, it is difficult to evaluate the model through the estimated coefficients [31]. Therefore, only the estimation results are presented in this study (e.g., Table 6), which will be subsequently analyzed, mainly through the generalized impulse response function and variance decomposition results of the PVAR models.

**Table 6.** PVAR Model GMM Parameter Estimates.

| Variables | TL | | | IND | | |
|---|---|---|---|---|---|---|
| | h_GNGT | h_CSU | h_TL | h_GNGT | h_CSU | h_IND |
| L.h_CNGT | 0.831 (−0.117) | −0.203 (−0.073) | 0.221 (−0.202) | 0.867 (−0.061) | −0.008 (−0.003) | −0.039 (−0.012) |
| L.h_CSU | 0.035 (−0.013) | 0.764 (−0.036) | 0.070 (−0.020) | 0.931 (−0.477) | 0.777 (−0.055) | −0.315 (−0.263) |
| L.h_TL | 0.033 (−0.056) | 0.030 (−0.028) | 0.823 (−0.104) | −0.092 (−0.187) | −0.002 (−0.014) | 0.858 (−0.089) |

Note: "h_" indicates the form that eliminates the sample individual/time fixed effects after the Helmert transformation; "L." indicates lag of order 1; standard errors in parentheses.

### 5.4. Impulse Response Function

The impulse response function plots for each variable lagged 20 periods are obtained using 1000 Monte Carlo simulations given unit standard positive difference shocks for each endogenous variable of the PVAR model, see Figures 1–6. Where the horizontal coordinate is the number of impulse response periods, the vertical coordinate is the degree of impact of the variable, the dashed middle line is the impulse response value, and the dashed lines on either side are the confidence interval lines at the 95% level.

### 5.4.1. Low-Carbon Green Transition, Consumption Upgrading, and Industrial Structure Rationalization

Figure 1 shows the pulse ringing diagram generated by itself, *CSU* and *TL* in the face of a forward unit standard deviation shock to *CNGT*. *CNGT* responds positively to its own shock, which peaks in the immediate period and then gradually decreases to zero and remains flat, suggesting that there is a clear 'locational advantage' to the low carbon green transition, and that its own level of improvement is closely linked to its past status, reflecting the continuity and cumulative nature of the low carbon green transition. The shock of *CNGT* causes a small negative response to *CSU* in the current period, which then decreases rapidly and peaks in period 2, before the negative response gradually decreases and eventually converges to zero; that is, *CNGT* has an enhanced dampening effect on *CSU* in the short term and is characterized by a long-term negative shock effect, which is generally consistent with the results of Zou and Wu [20]. This is mainly due to the fact that the low-carbon green transition of industries will have an impact on existing industries and accelerate the elimination of backward production capacity and the decline of some industries, which will not only cause an imbalance between product supply and consumption, but also tend to cause large fluctuations in employment and income, thus creating a dampening effect on consumption upgrading. In the face of a positive *CNGT* shock, *TL* generates a small negative value in the immediate period, then rapidly turns positive and reaches a maximum in period 3 before gradually converging to a small positive number, suggesting that low-carbon green transition inhibits the rationalization of industrial structure in the short term, while it has a catalytic effect in the long term. As emphasized by Fan and Liu [15], in the short term, higher environmental costs will squeeze investment in technological innovation, thus brings "Pains" of innovation as the industrial structure is rationalized and upgraded, while with the rapid release of dividends brought by low-carbon green transition, the low-carbon green transition can contribute to the upgrading of structural rationalization in the long term and continuously.

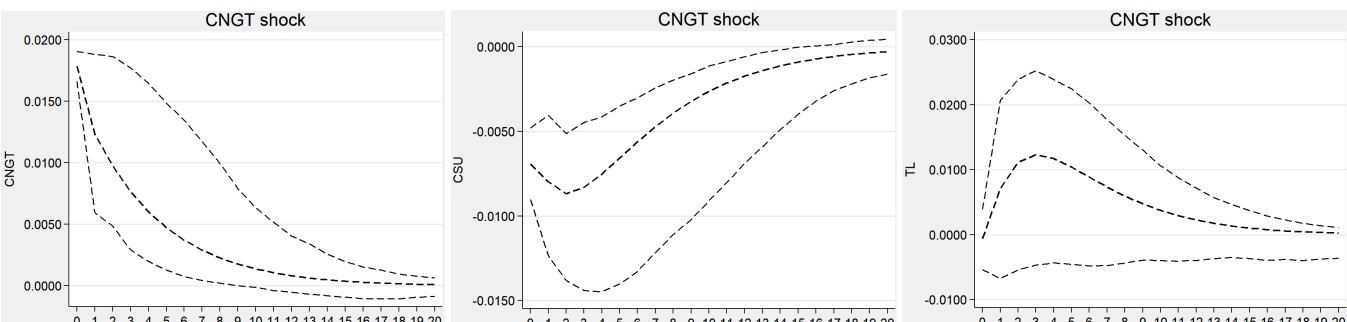

**Figure 1.** Impulse response function of *CNGT* to *CNGT*, *CSU* and *TL*.

Figure 2 shows the dynamic response of *CNGT*, *TL*, and itself to a positive unit standard deviation shock applied to the *CSU*. In the face of a *CNGT* unit standard deviation shock, *CNGT* has no response in the current period, and then the response value develops rapidly and positively, reaching a high point in periods 5–6 and then gradually declining and converging to zero. The response value is always positive, indicating that consumption upgrading will promote the low-carbon green transition but with a lag. This is mainly due to the constraints of cultural literacy, aesthetic preferences and understanding, residents often need a certain amount of time to change their existing consumption concepts and patterns, and thus consumption upgrades will lag behind the low-carbon green transition, while once consumption concepts and patterns are changed, residents' consumption potential can be effectively stimulated [3,4,14], thus forming a continuous promotion of the low-carbon green transition. The *TL*, on the other hand, generates a small negative value in the current period when faced with a *CNGT* shock, then the negative response weakens rapidly and turns positive in period 1, after which the positive response value becomes larger rapidly and gradually decreases and converges to zero after reaching a peak. This is

probably because the expansion and upgrading of consumption are ahead of the change in industrial structure; enterprises need to change their production mode to adapt to the trend of consumption upgrade, and with the rational allocation of production factors and the coordinated development of various industries, the role of consumption structure upgrading in promoting the rationalization of industrial structure will continue to emerge. With the rational allocation of production factors and the coordinated development of various industries, the role of the upgrading of the consumption structure in promoting the rationalization of the industrial structure continues to emerge and manifests itself as a long-term promotion [9,15]. In addition, as seen in Figure 2, consumption upgrading has a dynamic continuity, and the existing development characteristics will influence its future development level.

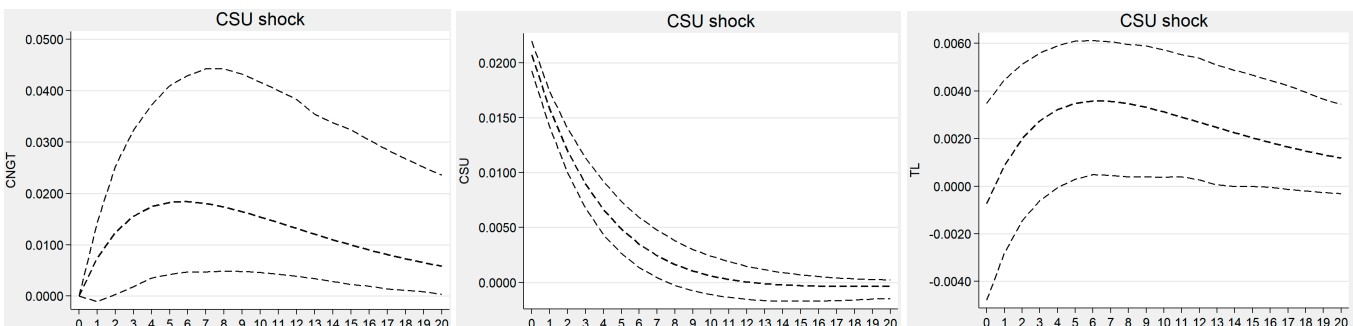

**Figure 2.** Impulse response function of *CSU* to *CNGT*, *CSU* and *TL*.

As seen in Figure 3, in the face of the positive unit standard deviation shock of *TL*, both *CNGT* and *CSU* do not respond in the current period. However, both show a weak positive increase and then gradually decrease and finally converge to zero, indicating that the rationalization of industrial structure may help promote low-carbon green transition and the upgrading of consumption. Even so, this effect is weak or even insignificant, which is largely consistent with the results of the Granger causality test. According to the study of Xie [9], if the demand for consumption upgrading can be met only through the transition of existing technological achievements, then producers can improve their own technological level and accelerate their productivity by improving their existing machinery and equipment and production processes, or by imitating the technology in competitors' or partners' products, which is undoubtedly the quickest way for enterprises to expand their production capacity and meet consumer demand. However, whether through transition or imitation, the technological achievements used do not break through the existing production frontier, so the rationalization of the industrial structure achieved in this way only follows the trend of consumer upgrading, does not significantly contribute to it, and for the same reason the low-carbon green transition cannot be significantly affected. In addition, similar to *CNGT* and *CSU*, may also have an "Incumbency Advantage" due to the dynamic continuity of its development.

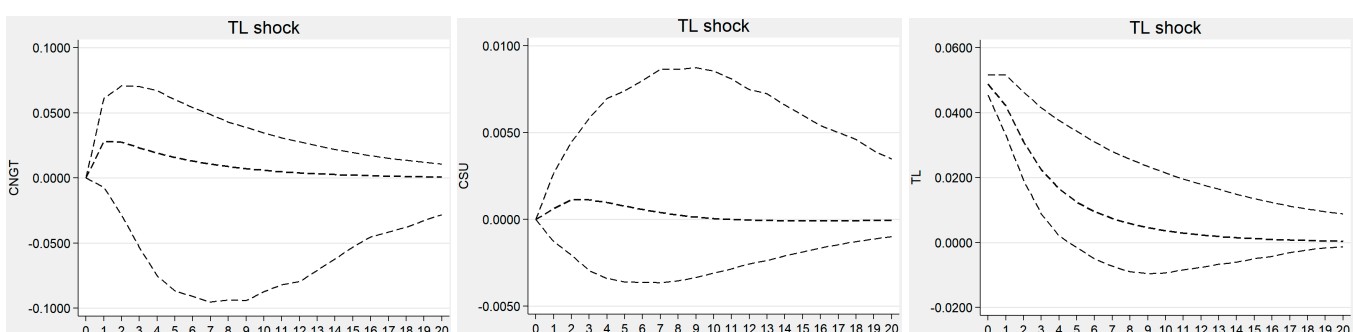

**Figure 3.** Impulse response function of *TL* to *CNGT*, *CSU* and *TL*.

### 5.4.2. Low-Carbon Green Transition, Consumption Upgrading, and Industrial Structure Upgrading

Figures 4–6 depict the overall changes in the relationship between the variables during the observation period from the perspective of industrial structure upgrading, in which the interaction between $CNGT$ and $CSU$ is basically consistent with the findings from the perspective of industrial structure rationalization, indicating that the estimation results of the model in this study have a strong robustness.

As seen in Figure 4, differing from $CNGT$ which continues to contribute positively to $TL$, $IND$ continues to respond negatively to the shock of $CNGT$, specifically by forming a small negative value in the immediate period, then dropping rapidly and reaching a peak in period 5, and then gradually approaching a value of zero, indicating that the low-carbon green transition is not conducive to promoting the industrial structure upgrading and that the impeding effect is persistent in the long term. The main reason for this is that, in the face of pressure for the low-carbon green transition, companies usually prioritize learning to improve their existing machinery and equipment in order to increase their output per unit of polluting emissions and thus their average lifespan, and that the adoption of environmentally friendly technologies often requires large capital investments, which constrains the incentive for technological innovation and transformation and hinders industrial upgrading of the whole economic system. The low-carbon green transition will also weaken the incentive of enterprises to first, research and develop technologies that help improve the efficiency of fossil energy and, second, stimulate investment in research and development (R&D) of alternative energy technologies. However, due to the scarcity of carbon emissions, the aforementioned behavior will restrict technology R&D activities, which is not conducive to industrial upgrading.

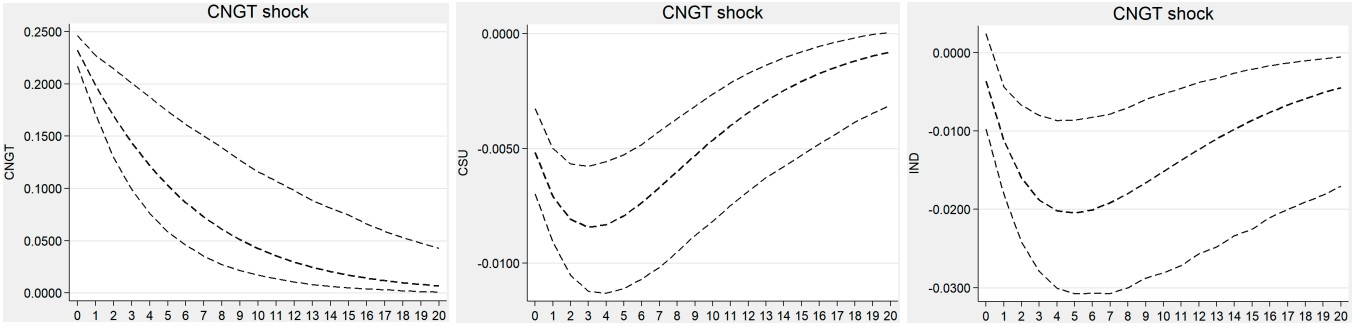

**Figure 4.** Impulse response function of $CNGT$ to $CNGT$, $CSU$, and $IND$.

According to Figure 5, the $IND$ responds more positively in the immediate period after a unit standard deviation shock to the $CSU$, followed by a gradual decline below the zero level and a convergence to zero from the negative direction. This suggests that consumption upgrading has an "Immediate" effect on industrial structural upgrading in the short term, with a more rapid but unsustainable impact; in the long term, consumption upgrading may not be conducive to the process of structural upgrading and has a continuous decaying effect, but the overall performance is positive promotion, which is basically consistent with the academic consensus. This may be due to the fact that in the process of consumption upgrading, in order to quickly obtain the dividends from the release of market potential, local governments tend to encourage and introduce technologies or enterprises with short cycles, low risks and quick results to upgrade the technological level of their own industries through policy tilts or financial support, and are prone to introduce similar industries at the same time, leading to a "Tidal wave" of industrial investment, resulting in the upgrading level of industrial structure. This will lead to a "Surge" of industrial investment and a significant increase in the level of industrial structure upgrading in the short term. However, the homogeneous development of industries, combined with the shortage of key core technology elements, will cause the problem of "Necking" when the industrial

structure is upgraded to a certain level. Thus, consumer upgrading is not conducive to promoting the industrial structure upgrading in the long term.

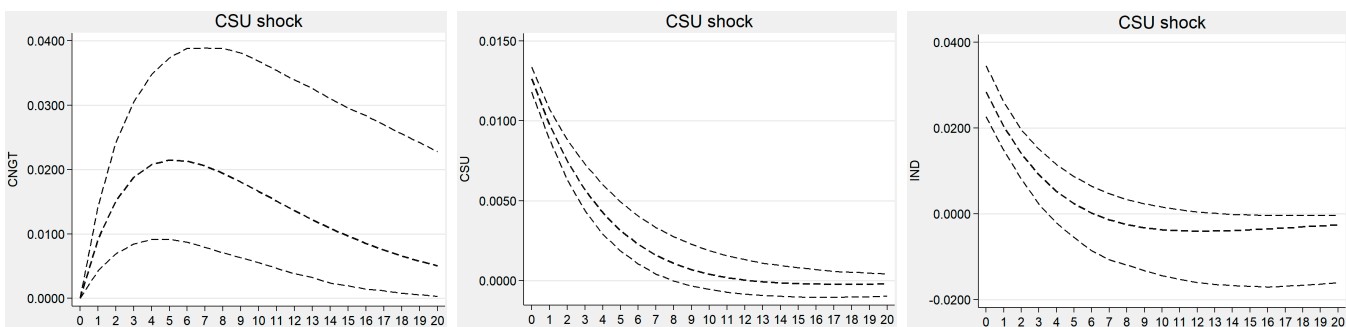

**Figure 5.** Impulse response function of *CSU* to *CNGT*, *CSU*, and *IND*.

According to Figure 6, the response of *CNGT* to the shock of *IND* is completely different from that of *TL*; that is, the industrial structure upgrading will have an enhanced inhibitory effect on the low-carbon green transition in the short term. The negative shock effect has long-term development characteristics, but the negative shock effect of the industrial structure upgrading also has a lag. Since the reform and opening up, China has relied on the competitive advantage of "Low price of resources and environment" and has laid out more industrial chain links with high consumption and high pollution. However, strategic new industries, such as clean energy conservation and high technology, have been expanded massively in recent years, and the constraints of industrial linkage have led to "High consumption" and "High emission" in the industrial chain. Although strategic new industries such as clean energy-saving and high-tech industries have been expanded on a large scale in recent years, due to the constraints of industrial linkages, the "High consumption" and "High emission" upstream and downstream links in the industrial chain continue to expand, thus forming a dilemma of reverse greening of the whole industrial chain. The response of *CSU* to the shock of *IND* has similar evolutionary characteristics to that of *TL*, implying that although the process of industrial structure upgrading lags behind the trend of consumption upgrading, it can better meet the personalized and diversified needs of consumers by promoting the gradual improvement of the quality of supply of products and services, stimulating consumption growth and sustainably promoting consumption upgrading in the long run.

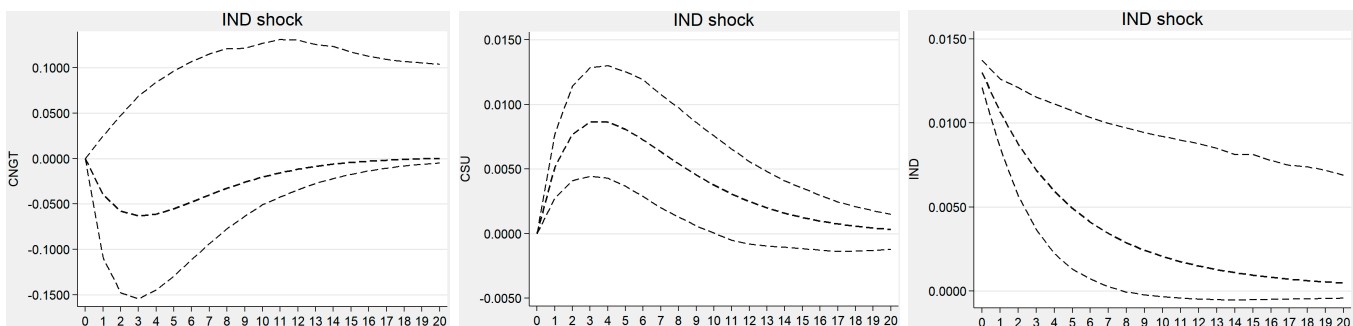

**Figure 6.** Impulse response function of *IND* to *CNGT*, *CSU* and *IND*.

### 5.5. Variance Decomposition

To further examine the strength of mutual explanations between variables, the study continues with variance decomposition to explain the relative importance of each indicator shock to the other variables. The results of the predicted variance decomposition for periods 5, 10, and 20 are shown in Table 7.

**Table 7.** Results of variance decomposition.

| Variables | Number of Issues | TL | | | IND | | |
|---|---|---|---|---|---|---|---|
| | | GNGT | CSU | TL | GNGT | CSU | IND |
| GNGT | 5 | 0.842 | 0.071 | 0.087 | 0.913 | 0.041 | 0.046 |
| GNGT | 10 | 0.757 | 0.110 | 0.134 | 0.811 | 0.087 | 0.102 |
| GNGT | 20 | 0.754 | 0.111 | 0.135 | 0.800 | 0.088 | 0.112 |
| CSU | 5 | 0.208 | 0.733 | 0.059 | 0.217 | 0.768 | 0.015 |
| CSU | 10 | 0.313 | 0.560 | 0.087 | 0.311 | 0.593 | 0.097 |
| CSU | 20 | 0.313 | 0.599 | 0.088 | 0.349 | 0.555 | 0.097 |
| TL/IND | 5 | 0.072 | 0.017 | 0.911 | 0.082 | 0.108 | 0.810 |
| TL/IND | 10 | 0.160 | 0.073 | 0.767 | 0.176 | 0.081 | 0.744 |
| TL/IND | 20 | 0.161 | 0.075 | 0.765 | 0.187 | 0.079 | 0.733 |

From the perspective of industrial structure rationalization, the contribution of low-carbon green transition to itself is the largest, with the contribution rate still reaching 75.38% in the 20th period; the contribution rates of consumption upgrading and industrial structure maintain an increasing trend, with 11.08% and 13.53%, respectively, in the 20th period, indicating that low-carbon green transition is more influenced by industrial structure rationalization than by consumption upgrading. The fluctuation of consumption upgrade is mainly influenced by itself and low-carbon green transition, of which 59.87% of the fluctuation can be explained by itself and 31.34% by low-carbon green transition, whereas the contribution rate of industrial structure rationalization to consumption upgrade is 8.79%. As for the rationalization of industrial structure, 76.47% of the fluctuations are explained by its own impact, whereas the contribution of low-carbon green transition and consumption upgrading to the rationalization of industrial structure is 16.8% and 7.46%, respectively, indicating that besides its own impact, low-carbon green transition and consumption upgrading also constitute important causes of fluctuations in the rationalization of industrial structure.

From the perspective of industrial structure upgrading, low-carbon green transition, consumption upgrading, and industrial structure upgrading fluctuations are mainly influenced by themselves, with their contribution rates still reaching 80.03%, 55.46%, and 73.34%, respectively, in the 20th period. The impact of low-carbon green transition on consumption upgrading and industrial structure upgrading is greater and maintains a growing trend, with the contribution rate of low-carbon green transition reaching 31.06% and 17.59%, respectively, in the 10th period. The contribution rate of consumption upgrading to low-carbon green transition has been on the rise, from 4.14% in Period 5 to 8.77% in Period 20, whereas the contribution rate of consumption upgrading to the industrial structure upgrading has been on a slow decline, from 10.8% in Period 5 to 7.94% in Period 20. The contribution of industrial restructuring to low-carbon green transition and consumption upgrading has continued to increase, with the contribution of industrial restructuring to both increasing from 4.56% and 1.54% in Period 5 to 11.2% and 9.68% in Period 20, respectively.

*5.6. Robustness Tests*

To ensure the robustness of the empirical results, the following methods are used to test the reliability of the empirical model: (1) the PVAR model is re-estimated by excluding some sample periods and intercepting the 2012–2019 panel data; (2) the developmental consumption and enjoyment consumption measured based on the consumption structure model are used as alternative indicators of consumption upgrading as proxy indicators for consumption upgrading and using total carbon emissions as a proxy indicator for green low-carbon transition for PVAR model analysis; (3) the consumption upgrading index is reconstructed using simple arithmetic mean and principal component analysis, and replacing the consumption upgrading index in the model.

The results of the impulse response function and variance decomposition based on the three aforementioned transformations show that, to some extent, some of the results differ from the benchmark results, but there is essentially no discrepancy with the conclusions of the benchmark model, indicating that the results of the benchmark model are robust and correctly reflect the intrinsic economic logic among the variables, and the results of the empirical analysis of the model are reasonably reliable. However, the results of the robustness estimation are not listed in this paper due to space constraints.

## 6. Research Conclusions and Policy Recommendations

### 6.1. Research Findings

By incorporating low-carbon green transition, consumption upgrading, and industrial structure change into a unified research framework, the interaction between low-carbon green transition, consumption upgrading, and industrial structure change is examined from two dimensions of industrial structure rationalization and upgrading by using the PVAR model based on provincial panel data from 2008 to 2020. (1) There is a significant two-way interaction between low-carbon green transition and consumption upgrading. In particular, whereas the low-carbon green transition has a persistent hindering effect on consumption upgrading, consumption upgrading helps promote the low-carbon green transition, but there is a time lag in the promotion effect of fee upgrading. (2) Heterogeneous interaction between different dimensions of low-carbon green transition and industrial structural change. The low-carbon green transition brings "Pain" to the rationalization of industrial structure in the immediate period but has a continuous facilitating effect afterward; the low-carbon green transition and the industrial structure upgrading show a continuous inhibiting effect on each other; that is, positive changes in either side are not conducive to the beneficial development of the other side. (3) There is a distinctly different interaction between consumption upgrading and industrial structure rationalization and industrial structure upgrading. After the immediate "Pain" consumption upgrading can give a lasting boost to the rationalization of the industrial structure; for the industrial structure upgrading, consumption upgrading has an "Immediate" boost in the short term; however, its impact is rapid but unsustainable, and in the long term, it has a certain hindering effect; the process of industrial upgrading lags behind the trend of consumption upgrading but can continue to contribute to consumption upgrading.

### 6.2. Policy Recommendations

Based on the aforementioned findings, this study proposes the following recommendations to better promote the construction of ecological civilization, promote the upgrading of consumption, accelerate the development of the modern industrial system, and achieve the integration and synergistic development of low-carbon green transition, consumption upgrading, and industrial structure change:

1. Continuously advocate the concept of green consumption and realize the integration and synergistic development of low-carbon and green transition and consumption upgrading. Improve low-carbon green environment-related laws and regulations and incentive policies, actively guide and encourage the development of good habits of ecological consumption according to the humanistic and economic realities of each region, reduce the waste of daily water and electricity resources, accelerate the laying of bus and metro lines around the city, encourage residents to travel green, reduce the emission of polluting gases, promote the formation of green low-carbon lifestyles and consumption patterns that are compatible with China's national conditions, and avert the dilemma of consumer expansion. This will help promote the formation of a green and low-carbon lifestyle and consumption pattern that aligns with China's national conditions, as well as overcome the interdependence between consumption expansion and environmental pollution.

2. Establish the policy orientation of the green whole industry chain and reshape the whole industry chain structure from the perspective of the green whole industry chain. To realize the green transition and upgrading of the whole industry, the construction of a green,

low-carbon circular economy system inevitably requires that industrial transformation and upgrading must be based on the greening of the whole industrial chain, establishing clean production and consumption patterns, improving the product "Ecological footprint" evaluation system, and formulating various technical policies, environmental standards, and industrial regulations corresponding to the green content of different links in a focused manner to cover any production process and effectively reduce the environmental impact of the whole industry chain, including the recycling process.

3. Consider the synergistic effect of imitation innovation and independent innovation to accelerate the pace of industrial transformation and upgrading. Policy design should pay close attention to global industrial technology trends, combined with its comparative advantages, and use introduced technology imitation innovation to improve its technology level and productivity. At the same time, the government should also anticipate potential directions that can be accepted by the dominant market design through R&D subsidies, tax breaks, and other preferential policies to stimulate local governments and enterprises to develop independent original key technology innovation, so as to break away from the "Path dependence" on technology pioneers, seize the high ground of industrial development, and take the lead in technology and environmental standards in the whole industrial chain.

There are some limitations in this study that require further improvement and expansion. First, this study does not analyze regional differences in the interaction between low-carbon green transition, consumption upgrading, and industrial structure change, which may have some impact on the conclusions and policy recommendations. Second, this study assumes that the regions are independent of each other and therefore does not consider the horizontal spatial influence between individual regions. Third, due to data constraints, this study does not conduct an in-depth exploration of the interaction between low-carbon green transition, consumption upgrading, and industrial structure change at the relatively micro level of cities and counties, which is a direction to be studied in the future.

**Author Contributions:** Conceptualization, X.X. and A.Y.; methodology, X.X. and A.Y.; software, X.X.; validation, X.X.; formal analysis, X.X.; investigation, X.X.; resources, X.X. and A.Y.; data curation, X.X.; writing—original draft preparation, X.X.; writing—review and editing, A.Y.; visualization, X.X.; supervision, A.Y. All authors have read and agreed to the published version of the manuscript.

**Funding:** This research received no external funding.

**Institutional Review Board Statement:** Not applicable.

**Informed Consent Statement:** Not applicable.

**Data Availability Statement:** The datasets used during the current study are available from the corresponding author on reasonable request.

**Conflicts of Interest:** The authors declare no conflict of interest.

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
