# Peer review of "Consumption Upgrading and Industrial Structural Change: A General Equilibrium Analysis and Empirical Test with Low-Carbon Green Transition Constraints"

_sustainability, doi:10.3390/su142013645_

Round 1
Reviewer 1 Report
Dear Authors I really appreciate your study, aas far as I am concerned, the work present an high scientific soundness, the construction of the model, despite the strong assumptions, takes into consideration and describes very carefully the different actors, the econometric analysis is consistent with the aims of the study and the results allow to developed many policy recommendation.
I have only some minor concern about:
I recommend doing two divided sections, one introduction and the other review of the literature.
In the introductory section it is necessary to better describe the aim of the study and to present the development of the other section of the paper.
In order to improve the quality of the conlusion section for other development:
See and Cite:
Lin, J. Y., & Wang, Y. (2020). Structural change, industrial upgrading, and middle-income trap. Journal of Industry, Competition and Trade, 20(2), 359-394.
Frittelli, M., Madzvamuse, A., & Sgura, I. (2021). Bulk-surface virtual element method for systems of PDEs in two-space dimensions. Numerische Mathematik, 147(2), 305-348.
Author Response
Point 1: I recommend doing two divided sections, one introduction and the other review of the literature.
Response 1: Accept the comments of the reviewer. In this paper, the original version of 'Introduction and Literature Review' is split into two parts: 'Introduction' and 'Literature Review', and the content and structure of these two parts have been reorganised to suit the needs of the paper.
Point 2: In the introductory section it is necessary to better describe the aim of the study and to present the development of the other section of the paper.
Response 2: Accept the comments of the reviewer. Changes made to the introduction section: (1) Restructured the introduction section to clarify the issues to be studied in the article, as detailed in the paper. (2) Additions related to the structural arrangement of the article in the introduction section.
Point 3: In order to improve the quality of the conlusion section for other development:
See and Cite:
Lin, J. Y., & Wang, Y. (2020). Structural change, industrial upgrading, and middle-income trap. Journal of Industry, Competition and Trade, 20(2), 359-394.
Frittelli, M., Madzvamuse, A., & Sgura, I. (2021). Bulk-surface virtual element method for systems of PDEs in two-space dimensions. Numerische Mathematik, 147(2), 305-348.
Response 3: Accept the comments of the reviewer. In response to the reviewer's comments, the literature review has been presented as a separate section, and the section has been re-found and adjusted according to the references provided by the reviewers. In addition, the marginal contribution of the article has been added to the literature review section. See the paper for details.

Reviewer 2 Report
Dated: 05-10-2022
Sustainability-1955719
Title: Consumption Upgrading and Industrial Structural Change: A general equilibrium analysis and empirical test with Low-carbon Green Transition constraints
Dear Sir,
The work is interesting, but there are some minor problems with this manuscript. The manuscript can be accepted for publication after minor revisions. Kindly see the comments list.
Remark: Accept after minor revision
List of comments
1. Introduction and Literature Review: Add the relevant citation/references in the following section.
2. Add the separate heading of conclusion with major conclusion.
3. Abstract part should be revised more critical.

Author Response
Point 1: Introduction and Literature Review: Add the relevant citation/references in the following section.
Response 1: Accept the comments of the reviewer. In this paper, the original version of 'Introduction and Literature Review' is split into two parts: 'Introduction' and 'Literature Review', and the content and structure of these two parts have been reorganised to suit the needs of the paper.
In the 'Introduction' section, we have further clarified the research questions of the paper and added relevant contents such as the structure of the article; in the 'Literature Review' section, We have re-found some of the literature and added to existing studies and have added marginal contributions to the article at the end of this section.
Point 2: Add the separate heading of conclusion with major conclusion.
Response 2: Accept the comments of the reviewer.Add the separate conclusion heading for the main conclusion in the conclusion section. See the paper for details.
Point 3: Abstract part should be revised more critical.
Response 3: Accept the comments of the reviewer.In response to the reviewers' comments, we have reviewed the abstract section and revised some of the content.

Reviewer 3 Report
I read and found that the manuscript has some merits. However, I have some comments.
1. The introduction section should clarify the contribution of this study. How is it different from existing studies on this topic? And, please discuss the topic as it relates to China.
2. Why is the PVAR method adopted? what is the advantage of this method? Please check for the condition for using the PVAR method and Granger causality method. Please provide a diagram of the econometric strategy.
3. The empirical result section is appreciated. However, the authors ignored the discussion. So, the readers will ask how is it different from previous studies. And, please explain the reason supporting your findings
4. Please provide your study limitation and further study.
Good luck with the revision.
Author Response
Point 1: The introduction section should clarify the contribution of this study. How is it different from existing studies on this topic? And, please discuss the topic as it relates to China.
Response 1: Accept the comments of the reviewer. In this paper, the original version of 'Introduction and Literature Review' is split into two parts: 'Introduction' and 'Literature Review', and the content and structure of these two parts have been reorganised to suit the needs of the paper.
In the 'Introduction' section, we have further clarified the research questions of the thesis ,and further specified that the research on the relevant issues is mainly based on the context of China. In addition, relevant elements such as the structure of the article have been added to the introduction section.
In the 'Literature Review' section, we have re-found some of the literature and added relevant content based on existing studies, and the marginal contribution of the article is described at the end of this section.
Point 2: Why is the PVAR method adopted? what is the advantage of this method? Please check for the condition for using the PVAR method and Granger causality method. Please provide a diagram of the econometric strategy.
Response 2: Accept the comments of the reviewer.
Based on the reviewers' comments, we have re-checked the conditions for the applicability of the PVAR model and the Granger causality test. In order to ensure the applicability of the PVAR model model, we re-specify the research question of this article in the introduction section, clearly stating that we want to investigate the interaction effects between low carbon green transformation, consumption upgrading and industrial structure change. In the literature review section, it is also stated again that the paper aims to investigate the mechanism of interaction between the three, and it is pointed out that the PVAR model, which can reveal the interaction effect of variables, can help to analyse the dynamic relationship between the three. The advantages of the PVAR model are explained in the empirical design section.
As the diagram was not uploaded correctly there, diagram of the econometric strategy is detailed in the Annex.
Point 3: The empirical result section is appreciated. However, the authors ignored the discussion. So, the readers will ask how is it different from previous studies. And, please explain the reason supporting your findings.
Response 3: Accept the comments of the reviewer. Considering that the results of the PVAR model mainly analyze the relationship between variables through the impulse corresponding function, for this reason we mainly discuss the relationship between the variables in this paper, and explain the reason that can support our findings.See the paper for details.
Point 4: Please provide your study limitation and further study.
Response 4: Accept the comments of the reviewer. At the end of Part VI “Research Conclusions and Policy Recommendations”, the paper points out the limitations in this study that require further improvement and expansion. See the paper for details.
